# Tunica intima compensation for reduced stiffness of the tunica media in aging renal arteries as measured with scanning acoustic microscopy

**Katsutoshi Miura** *

Department of Health Science, Pathology and Anatomy, Hamamatsu University School of Medicine, Hamamatsu, Japan

* kmiura.hama.med@gmail.com

## Abstract

### Objectives

Aging causes stiffness and decreased function of the renal artery (RA). Histological study with light microscopy can reveal microscopic structural remodeling but no functional changes. The present study aimed to clarify the association between structural and functional aging of the RA through the use of scanning acoustic microscopy.

### Methods

Formalin-fixed, paraffin-embedded cross-sections of renal arteries from 64 autopsy cases were examined. Speed-of-sound (SOS) values of three layers, which correspond to the stiffness, were compared among different age groups. SOS of the tunica media was examined in terms of blood pressure (BP) and SOS of the ascending aorta. Vulnerability to proteases was assessed by SOS reduction after collagenase treatment.

### Results

The tunica intima presented inward hypertrophy with luminal narrowing, and the tunica media showed outward hypertrophic remodeling with aging. SOS of the tunica media and internal and external elastic laminae showed a reverse correlation with age. SOS of the tunica media was negatively correlated with BP and strongly associated with that of the aorta. The tunica media of young RAs were more sensitive to collagenase compared with the old ones.

### Conclusions

Scanning acoustic microscopy is useful for observing the aging process of the RA. This technique simultaneously shows structural and mechanical information from each portion of the RA. In the process of aging, the RA loses contractile function and elasticity as a result of protease digestion. The tunica media and the internal and external elastic laminae exhibit reduced stiffness, but the tunica intima stiffens with atherosclerosis. As a consequence, the RA's outer shape changes from round to oval with inward and outward hypertrophy. This

**Data Availability Statement:** All relevant data are within the paper and its Supporting Information files.

**Funding:** This study was funded by the Japan Society for the Promotion of Science KAKENHI (grant no. 17K1088901 and 15K08375), and by the Toukai Foundation for Technology. The funders had no role in study design, data collection and analysis, decision to publish, or preparation of the manuscript.

**Competing interests:** The author has declared that no competing interests exist.

indicates that the inner resistant intima supports the mechanical weakness of the tunica media to compensate for an increase in BP with aging.

## Introduction

Arterial stiffness is the consequence of structural and functional changes of the vascular wall that occur in response to cardiorenal metabolic syndrome, injury, or aging [1].

Measurements of functional change in renal arteries (RAs) include central pulse pressure/ stroke volume index, pulse wave velocity (PWV), total arterial compliance, pulse pressure (PP), and augmentation index [2]. PWV and PP, in particular, are two significant indices of arterial stiffness [3] that typically increase with age [4].

Conventional imaging methods involved in the diagnostic modalities of RAs are Doppler ultrasonography, scintigraphy, computed tomographic angiography, magnetic resonance arteriography, and angiography [5,6]. However, only histological study with light microscopy (LM) can reveal microscopic structural changes. LM information contributes to the detection of morphological alteration but not functional changes such as tissue stiffness or fragility. Although aging RAs usually show severe atherosclerosis and calcification, the aging process of other layers, such as tunica media, is not well known. Smooth muscles may lose contractile function and corresponding structural changes may occur with aging.

The aging process is commonly associated with increased vascular rigidity and decreased vascular compliance [5]. Although several researchers have reported functional or structural changes associated with aging [1,7], simultaneous acquisition of structural and functional information with the same histological specimen is rare. If histological and functional information can be obtained simultaneously from the same slide, the association of mechanical and the corresponding structural changes in the aging process may be more understandable and precise for comparing lesions.

Scanning acoustic microscopy (SAM) reveals morphological information and mechanical properties because speed-of-sound (SOS) through tissues corresponds to the stiffness of the content [8–11]. Simultaneous detection of histological structure and the corresponding tissue stiffness can help characterize mechanical weakness or strength of each RA element, including the tunica intima, tunica media, tunica adventitia, and internal and external elastic laminae (IEL and EEL, respectively). Moreover, LM shows analog images, whereas SOS initially presents digital images, thereby rendering it easy to statistically compare the differences. Moreover, by protease incubation of the section, the vulnerability of tissue components to enzymatic digestion is comparable among different specimens.

The present study aimed to use SAM observation to clarify the association of mechanical and structural changes in RAs during aging. All sample specimens were human RAs from autopsy cases of Japanese individuals of different ages. Stiffness, blood pressure, thickness of each layer, inner and outer diameters, and sensitivity to collagenase digestion were compared, and aging progression was summarized in a schematic image.

## Materials and methods

### Subjects and ethics

The study protocol conformed to the ethical guidelines of the Declaration of Helsinki and was approved by the ethical committee of the Hamamatsu University School of Medicine (approval no. 19–180). Because the study used stored autopsy samples without a link to the patient

identity, the need for written consent was waived. All procedures were conducted according to approved guidelines and regulations of the Ethic Committee.

All RAs and ascending thoracic aortae were obtained from autopsy cases of the Hamamatsu University Hospital in Japan. In total, 64 adult cadavers (48 men, 16 women) (S1 Table) without severe cardiovascular diseases, transplantation, or hemodialysis were consecutively selected (cause of death: neoplasm in 40, inflammation in 11, circulation disorders in 4, metabolic disorders in 4, and others in 5) to investigate the effects of aging. Their ages ranged from 16 to 101 years, and the mean age was 62.9 years (standard deviation, 16.3 years). RAs at the kidney hilum were cut into round slices, and ascending aortae were dissected into longitudinal sections. When accessory or aberrant RAs were present, the main or normal RA was selected for the study. Formalin-fixed, paraffin-embedded tissue blocks were flat-sectioned into 10-μm-thick slices and observed with SAM. Massively calcified tissues were decalcified via soaking in a mixture of formic acid and hydrochloric acid. Sections with focal defects or uneven surfaces were omitted for measurement. Clinical data, including the cause of death and, mean blood pressure (MBP) during life, were obtained from the clinical records.

## Scanning acoustic microscopic observations

We evaluated renal specimens using a SAM system (AMS-50AI; Honda Electronics, Toyohashi, Aichi, Japan) with a central frequency of 320 MHz and a lateral resolution of 3.8 μm [10]. The transducer was excited with a 2-ns electrical pulse to emit an acoustic pulse [12]. Samples were placed on the transducer, and distilled water was used for coupling fluid between the transducer and specimen. The transducer was used for both transmitting and receiving the signal. Waveforms reflected from the surface and bottom of the sample were compared to measure SOS and thickness of each point. The waveform from a glass surface without the sample served as the reference, with SOS only through water; 1495 m/s was used as a standard value.

The specimen was observed via the same method reported previously [10,13]. For fresh frozen specimen, the specimen frozen in n-hexan in -80˚C was cut in 10-μm thickness. The section was fixed in 95% ethanol or 10% buffered formalin before observation. To examine fixation effects, the sections were fixed in ethanol for 20 seconds and 10min and the same section was additionally fixed in formalin for 3 min and 1 hour. For formalin-fixed, paraffin-embedded (FFPE) specimens,

the 10-μm-thick specimen was dewaxed in xylene, gradually soaked in ethanol at decreasing concentrations, and finally washed in distilled water, and placed on the transducer. Both nearby sections were prepared to determine the differences between FFPE images and fresh frozen images. The mechanical scanner was arranged so that the ultrasonic beam was transmitted over the specimen to provide SOS values from each point. One cross-section for each person was measured. The number of sampling points was 300 in one scanning line, and each frame comprised $300 \times 300$ points. Mean SOS values were calculated from the values of eight different areas of each layer of the RA. The points of interest were randomly selected from cross points on the lattice screen [9]. The length in SOS images, typically in an area of 1.2 or 0.6 mm$^2$, was shown on the basal horizontal or lateral left bars on the screen.

Although FFPE specimens showed higher SOS than the fresh ones, SOS values were stable [14] (S1 File) irrespective of the length of formalin fixation (from 1 day to 3 months) [15]. Therefore, sample bias resulting from the fixation condition was negligible. Areas without calcified deposits and heavy atheroma were selected for comparison because calcified areas result in chatter marks on the section, causing irregular reflection, and large atheromatous portions become translucent as a result of lipid dissolution in organic solvents.

### Light microscopic observation

The same or nearby sections of SAM were stained with hematoxylin and eosin (HE) and with Verhoeff's elastic and Masson's trichrome (EMT) stain for comparison. Using EMT staining, the collagen and elastic fibers were stained blue and black, respectively.

### Measurement of the length of each layer and the longest and shortest of tunica media

The length of each layer was measured from LM images. The mean length was calculated from at least four points of the layer. The outer and inner axes of the tunica media were measured using LM images. The median length of the longest and shortest distances of each axis was assessed as the lengths of the outer and inner axes, respectively.

### Catalytic damage according to collagenase digestion

RAs from 3 young cadavers (40-, 44-, and 45-year-old men; "young RAs") and 3 old cadavers (82- and 90-year-old men and an 81-year-old woman; "old RAs") were selected for comparison. Paraffin sections were dewaxed with xylene, soaked in distilled water, and submerged into a solution of phosphate-buffered saline containing 0.5 mM of calcium chloride (pH 7.4) and 250 units/mL type III collagenase (Worthington, Lakewood, NJ, USA) at 37°C for 1.5 h or for 3h [16]. The collagenase used has substrate specificity to collagens with lower proteolytic activity than other collagenases. Digested sections were first washed with distilled water before being observed with SAM. The same sections were measured 1.5 h and 3 h after digestion.

### Statistical analyses

Mean SOS values were calculated from at least eight different areas per layer. The SAM manufacturer's software (LavView 2012, National Instruments, Austin, TX, USA) and commercial statistics software (Statcel3 add-in forms on Excel; OMS Publishing, Tokorozawa, Saitama, Japan), which calculated the mean areas-of-interest values, were used. Scatterplots showing the correlations between age and SOS values were established and subjected to simple linear regression analysis. The correlation strength was quantified with Pearson's correlation coefficients *(r)*.

Mean SOS values following collagenase treatment were compared at 0, 1.5, and 3 h using paired *t*-test.

One-way analysis of variance was used to compare SOS values among the different age groups and sections in different fixatives and under FFPE conditions. Multiple comparisons were assessed using Tukey–Kramer test.

Before the statistical analyses were conducted, all data sets that exhibited a normal distribution were compared in a test for the difference between mean values. A *P* value of $<0.05$ was considered statistically significant for all analyses.

## Results

### Differences in SOS images among fresh-frozen and FFPE sections

SOS images were compared among fresh frozen and FFPE sections from the same specimen to investigate the effects of fixation and paraffin-embedding on SOS values (Fig 1). No significant differences in SOS values was found between the FFPE and fresh-frozen sections in different fixatives (Fig 2 and S2 Table).

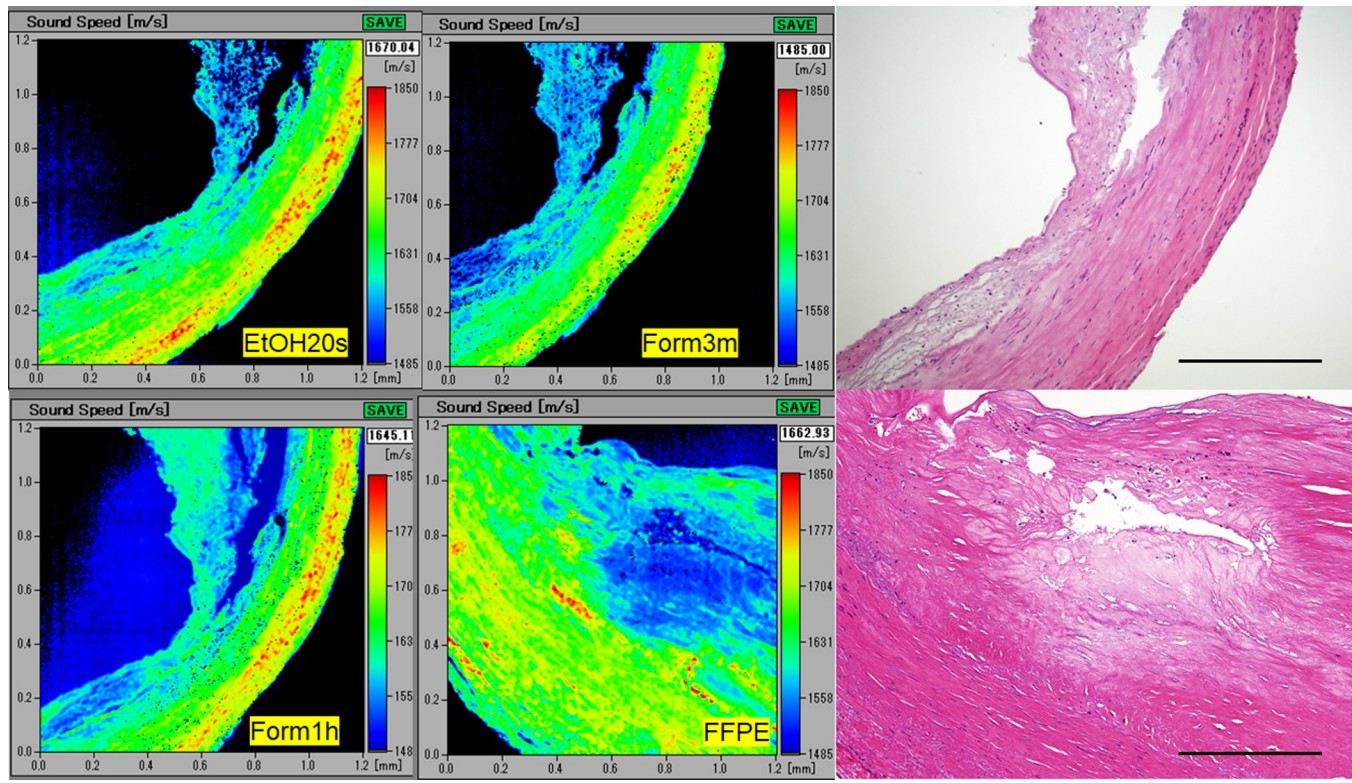

**Fig 1. SOS images of fresh-frozen and FFPE sections of RA.** The fresh-frozen sections in ethanol and formalin fixative, and FFPE sections showed similar SOS images. The intimal layer with high lipid content presented lower SOS areas, while the medial layer, especially the outer side with rich smooth muscle, demonstrated higher SOS areas. Furthermore, the HE images showed atheroma deposition in the intima. The scale bars represent 500 μm. Form, formalin.

## Speed-of-sound images and their corresponding light microscopic images associated with aging

The outer shape of the young RAs was round, and with age, it gradually changed to irregular oval with dilatation (Fig 3 top). The three-layered structure comprised the tunica intima, tunica media, and tunica adventitia in young RAs, the differentiation of which progressively became obscure in middle-aged and old RAs, where IEL and EEL tended to be thinner or to disappear. Hypertrophy of the tunica intima with atherosclerosis progressed with aging.

In SOS images (Fig 3 middle and bottom), IEL and EEL showed evidently high SOS, and the tunica media was positioned between them. These laminae were thick and continuous in young RAs and tended to be thinner and interrupted in old RAs. The tunica media was composed of thick, condensed smooth muscles with high SOS in young RAs and gradually changed to lean, sparse smooth muscles with low SOS in old RAs. Tunica intima accumulated atheromatous lipid material with low SOS values. Tunica adventitia, mainly composed of collagen bundles, displayed no remarkable changes in SOS with aging.

Fig 4A is a scatterplot of age and mean SOS values of the middle RA layer (S3 Table). The individual dot represents the mean SOS of each person. Linear regression fit showed a significant inverse correlation between age and SOS of the tunica media ($n = 64$, $r = -0.37$, $P = 0.0027$).

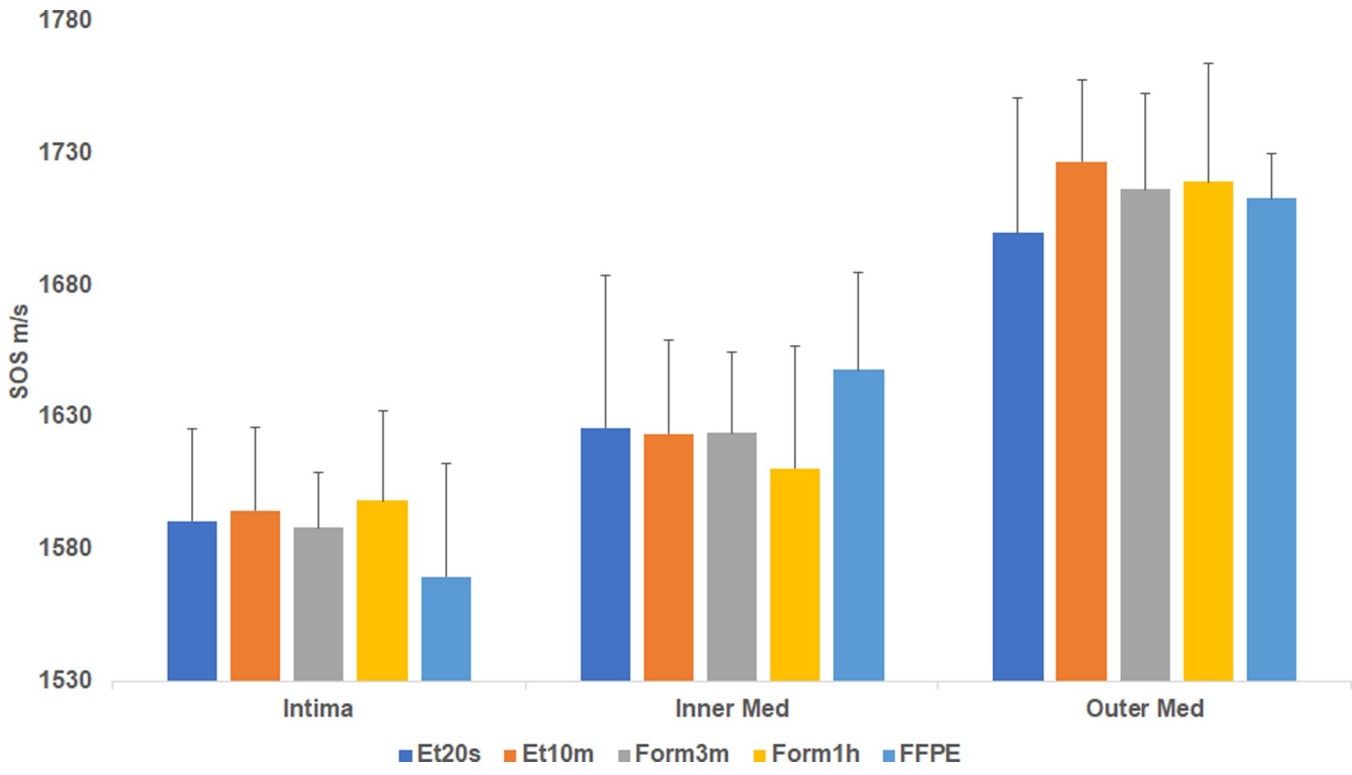

**Fig 2. SOS values (mean ± SD) of fresh-frozen sections in different fixatives and FFPE sections.** The fresh-frozen section fixed in 95% ethanol for 20 sec or 10 minutes, and the same section additionally fixed in 10% formalin for 3 minutes and 1 h, were compared to the nearby FFPE section. No significant differences were found either in the intimal and medial layers among the different groups. ET20s: ethanol 20 seconds, Form3m: formalin 3min, Inner Med: inner medial layer, Outer Med: outer medial layer.

### Speed-of-sound values of internal and external elastic laminae associated with aging

SOS of IEL and EEL significantly decreased with aging (Fig 4B and S4 Table). Both SOS values presented reverse correlation with age (n = 36, IEL: $r = -0.369$, $P = 0.0026$; EEL: $r = -0.643$, $P = 0.000000183$).

### Relationship between speed-of-sound values of tunica media and mean blood pressure

Fig 5A and S5 Table show the inverse correlation between MBP and SOS values of the tunica media ($n = 37$, $r = -0.37$, $P = 0.0019$). In RAs from cadavers who had had higher BP, SOS values in the tunica media were lower. Systolic and diastolic pressures and PP significantly increased with age (Fig 5B; for systolic pressure, n = 37, $r = 0.44$, $P = 0.00018$; for diastolic pressure, n = 37, $r = 0.36$, $P = 0.0024$; and for PP, $r = 0.32$, $P = 0.0076$) (S6 Table).

### Thickness of the three layers associated with aging

Fig 6A and S7 Table depicts the alteration in the width of each segment of RAs with aging. The widths of the tunica media and tunica adventitia showed no remarkable changes, whereas tunica intima exhibited remarkably increased thickness with aging (n = 35, $r = 0.43$, $P = 0.0002$).

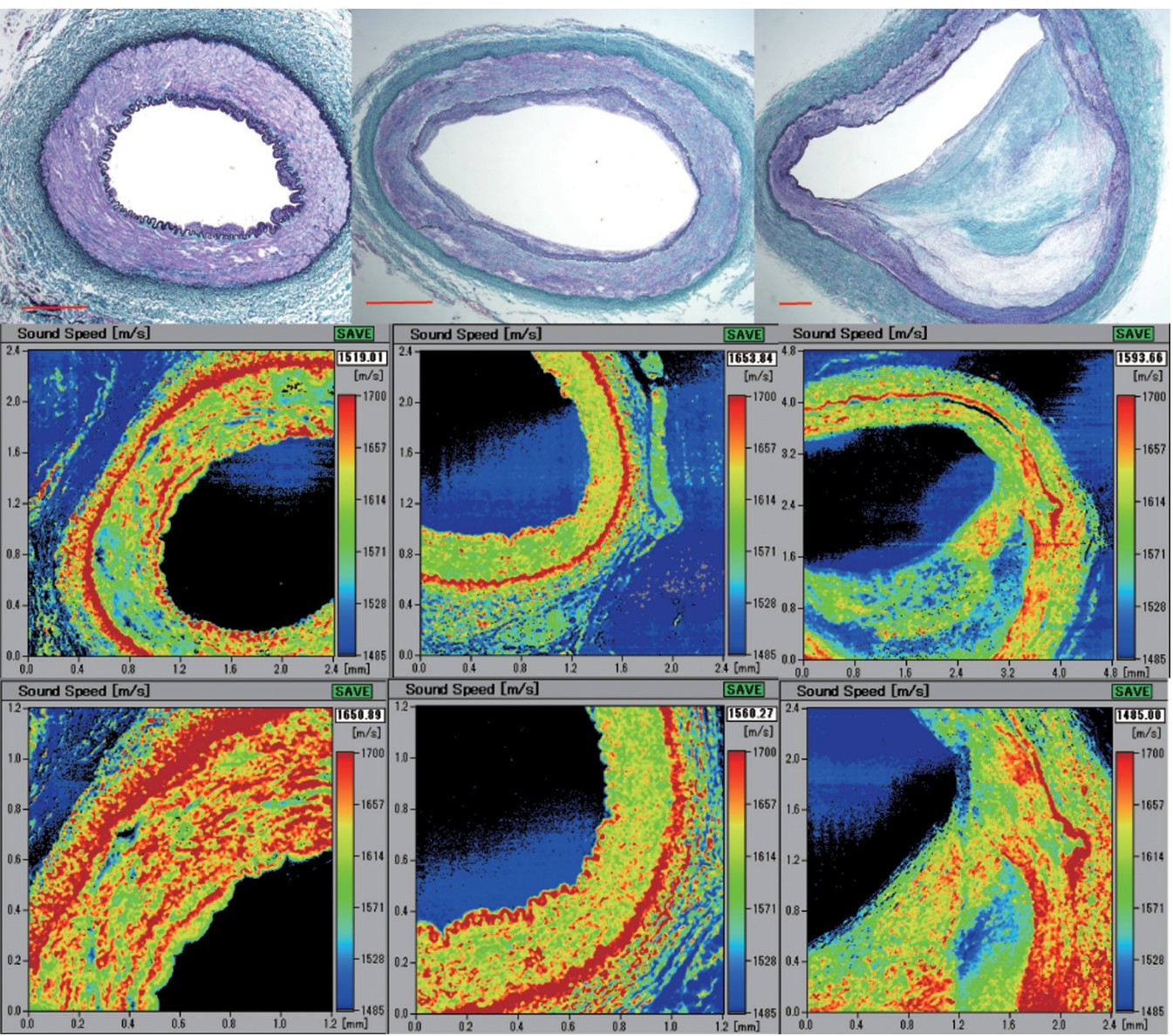

**Fig 3. Light microscopic images of the renal artery (RA) in Verhoeff's elastic and Masson's trichrome (EMT) stain and their corresponding speed-of-sound (SOS) images.** Top row: EMT-stained images. Left: RA from a cadaver in its 30s. Middle: RA from a cadaver in its 50s. Right lane: RA from a cadaver in its 80s. Middle row: low-magnification SOS image. Bottom row: high-magnification SOS image. The scale bars represent 500 μm.

## Thickness of the outer and inner mean axes of the tunica media

Although the width of the tunica media was rather stable with aging, the outer and inner mean axes of the tunica media gradually expanded with age (Fig 6B; for the outer axis, n = 35, r = 0.332, P = 0.0068; for the inner axis, n = 35, r = 0.511, P = 0.000133) (S8 Table). As a consequence, RAs became dilated with aging.

## Comparison between speed-of-sound values of renal artery and aorta

The alteration in SOS values in the tunica media of RAs was compared with that of the ascending aorta (Fig 7A). Both SOS values progressively increased with aging. SOS of the RA was

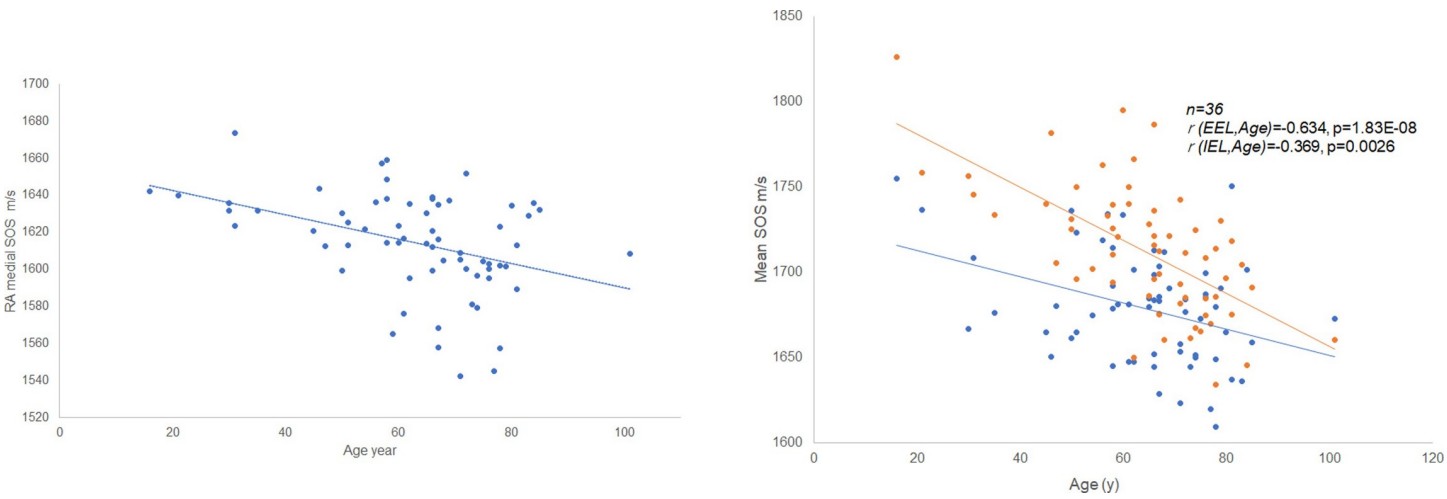

**Fig 4. Relationship between age and speed-of-sound (SOS) values.** (a) Scatterplot of age and average SOS values of the tunica media of the renal artery (RA). The individual dot represents the average SOS of each person. Linear regression fit showed a statistically significant inverse correlation between age and SOS of RA ($n = 64$, $r = -0.37$, $P = 0.0027$). (b) Speed-of-sound values of internal (IEL) and external (EEL) elastic laminae associated with aging. Both SOS values demonstrated reverse correlations with age ($n = 36$, for IEL, $r = -0.369$, $P = 0.0026$; for EEL, $r = -0.643$, $P = 0.000000183$).

always lower than that of the aorta, and the correlation was positive (Fig 7B; $n = 33$, $r = 0.51$, $P = 0.000146$) (S9 Table).

### Difference in sensitivity to collagenase treatment between young and old renal arteries

Owing to collagenase treatment, SOS values in the tunica media progressively decreased in young RAs, whereas they remained stable in old RAs (Fig 8 and S10 Table). Young tunica media with rich smooth muscles and elastic fibers showed reduced SOS values. In contrast, old tunica media with abundant collagen fibers and reduced smooth muscles maintained stable SOS values during digestion, particularly the inner side near intima. Statistical analysis showed a significant decline in SOS values in young RAs compared with old RAs (Fig 9).

### Summary of RA alteration in structure and fragility associated with aging

Fig 10 shows the schematic image changes in RAs associated with aging. Young RAs had round, regular three-layered structure with continuous, thick EEL and IEL. Old RAs had an irregular oval structure with hypertrophy of the tunica intima due to atherosclerosis. Tunica media of old RAs exhibited a decrease in smooth muscles and had elastic fiber splitting. In terms of mechanical strength, young RAs showed high SOS values in the tunica media, EEL, and IEL, indicating the high stiffness of these structures. In contrast, old RAs revealed irregularly low SOS values in the tunica media, EEL, and IEL, indicating the loss of mechanical strength in these structures.

## Discussion

The SOS values measured by SAM were not directly relevant to the flexural mechanical properties. However, SOS and soft tissue elasticity have a positive correlation, as shown in the

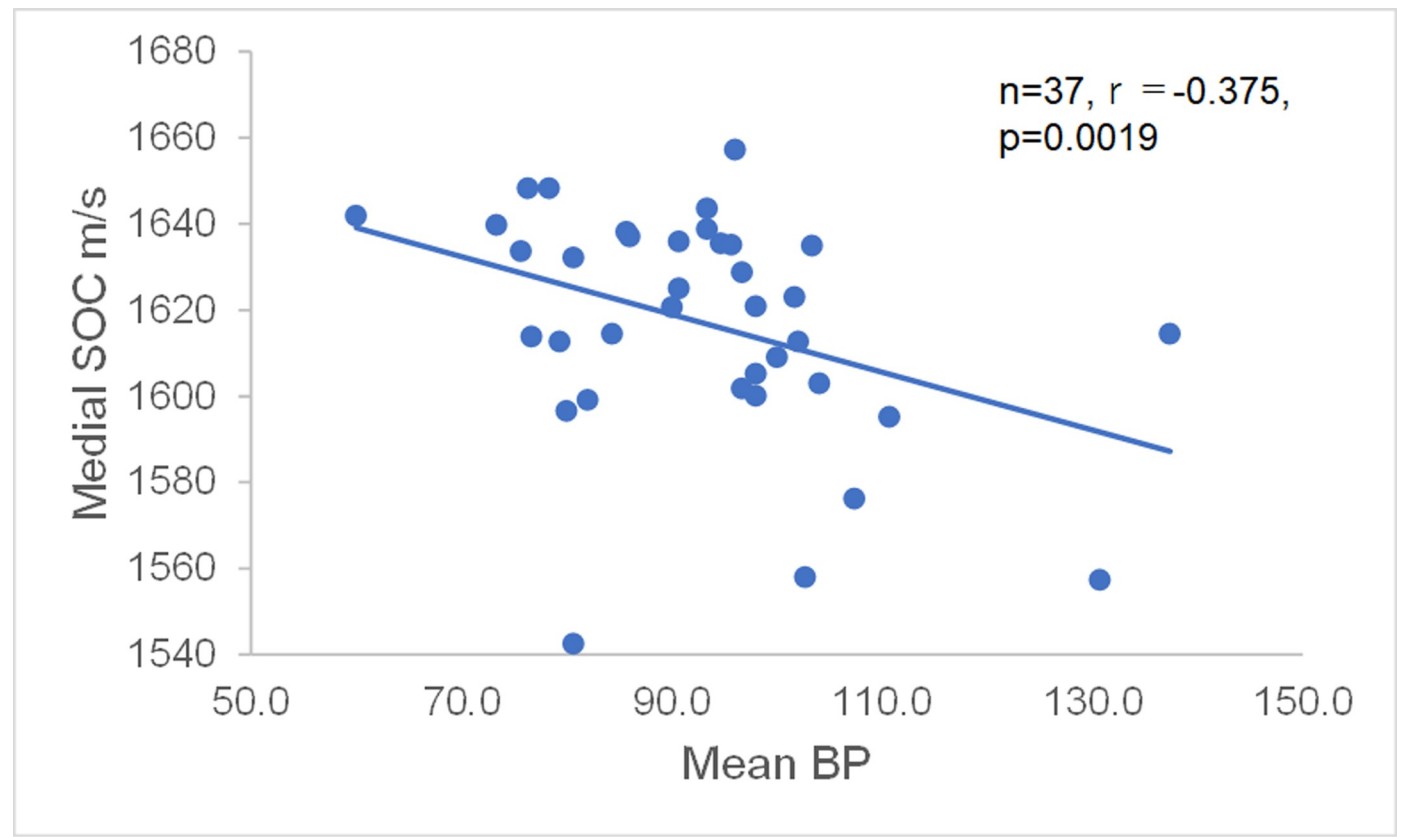

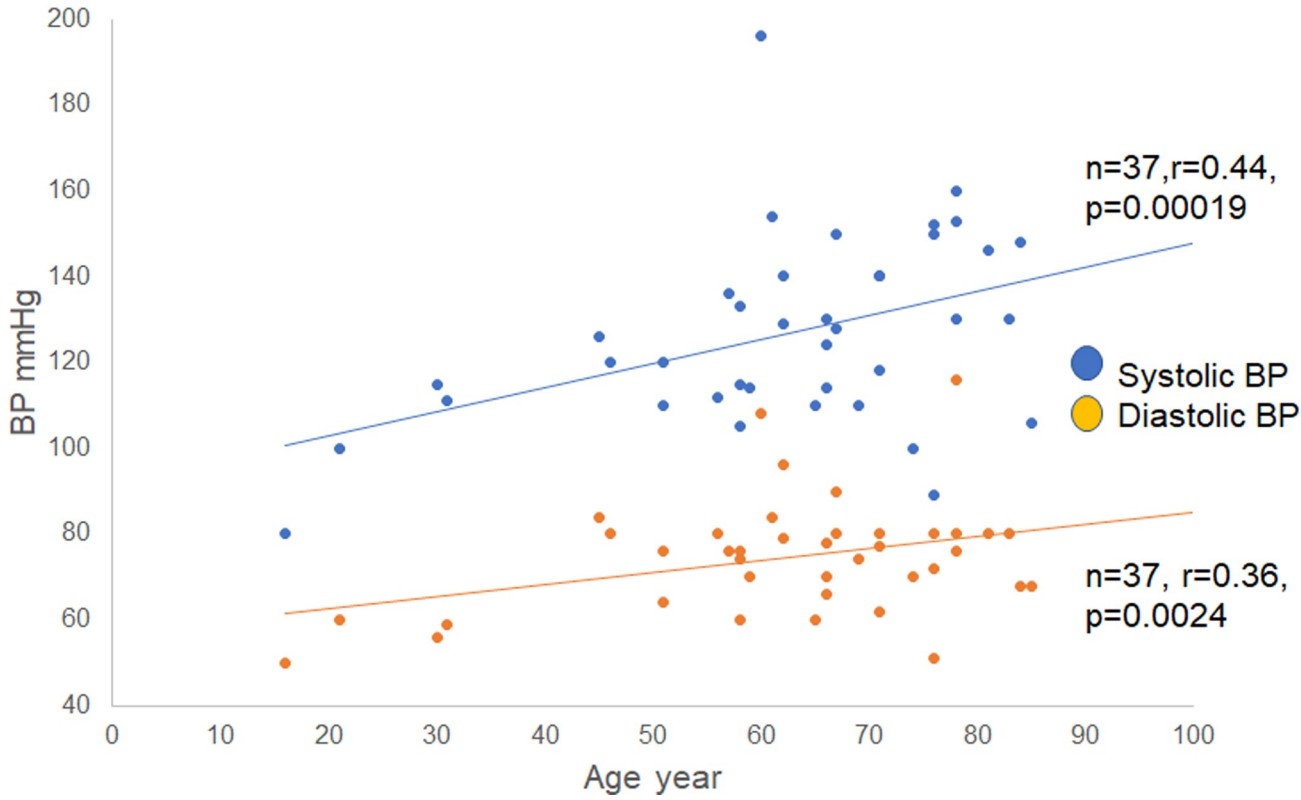

**Fig 5. Relationship between age and blood pressure.** (a) Inverse correlation between mean blood pressure (MBP) and speed-of-sound (SOS) values of tunica media. In renal arteries (RAs) with higher blood pressure, SOS values in the tunica media were lower. (b) Systolic and diastolic blood pressure associated with aging. Systolic and diastolic pressor pressures increased significantly with age (for systolic pressure, r = 0.44, P = 0.00018; for diastolic pressure, r = 0.36, P = 0.0024). The pulse pressure also increased significantly with aging (r = 0.32, P = 0.0076).

following equation [8, 17]:

$$c = (K/\rho)^{1/2}$$

where c = speed of sound, K = elastic bulk modulus, and ρ = tissue density.

For a homogeneous isotropic solid material, the ratio of stress (force per unit area) /strain (proportional deformation) is a constant and is called the modulus of elasticity. Young's modulus (longitudinal elasticity), shear or torsion modulus (rigidity), and bulk modulus (volume elasticity) are commonly used to compare elasticity in biological tissues. When a material is stressed, its breadth may contract as its length extends. This is defined by a constant called Poisson's ratio, which is expressed as

$$\sigma = (\text{lateral contraction per unit breadth})/(\text{longitudinal extension per unit length}).$$

Poisson's ratio is usually between 0.490 and 0.499 for soft tissues which are almost incompressible [17].

There are 3 constitutive equations to define the relationship between 3 elastic modules.

$$G = E / (2(1 + \sigma))$$

$$\sigma = (E / 2G) - 1$$

$$K = E / (3(1 - 2\sigma))$$

Where G = shear modulus, E = Young's modulus, σ = Poisson's ratio, and K = bulk modulus.

Soft tissues such as RAs are not homogeneous isotropic materials but heterogenous anisotropic materials with elastic and viscous properties that show a variable stress-strain response.

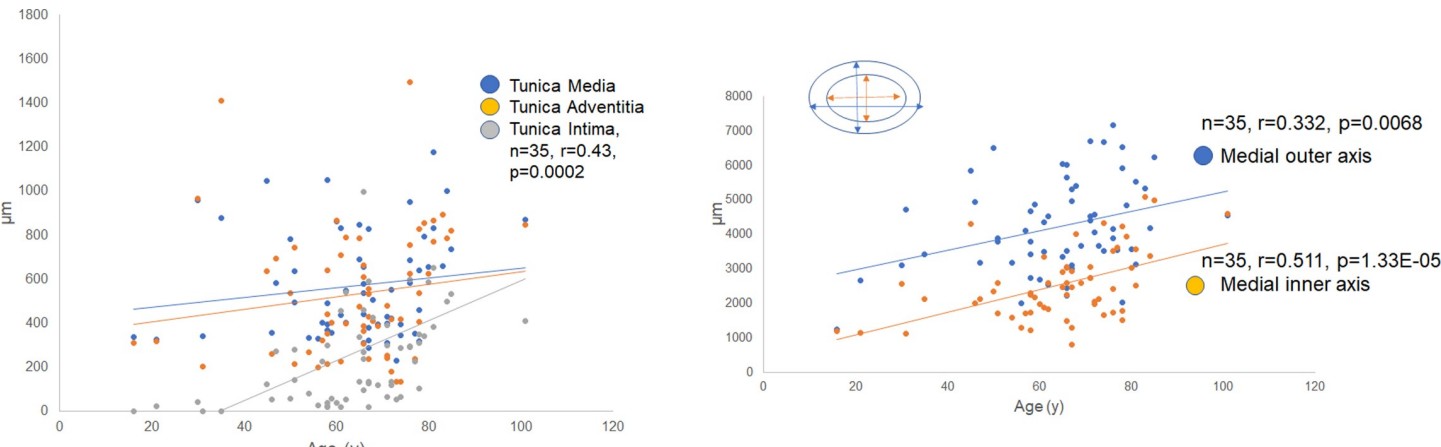

**Fig 6. Thickness of each layer of the renal artery (RA) with aging.** (a) The width of the three layers with aging. The thickness of the tunica media and tunica adventitia showed no remarkable changes, whereas the tunica intima exhibited increased thickness with aging (n = 35, r = 0.43, P = 0.0002). (b) The outer and inner axes of tunica media with aging. Both the outer and inner axes of the tunica media significantly increased with aging. As a consequence, the renal artery (RA) became dilated with aging.

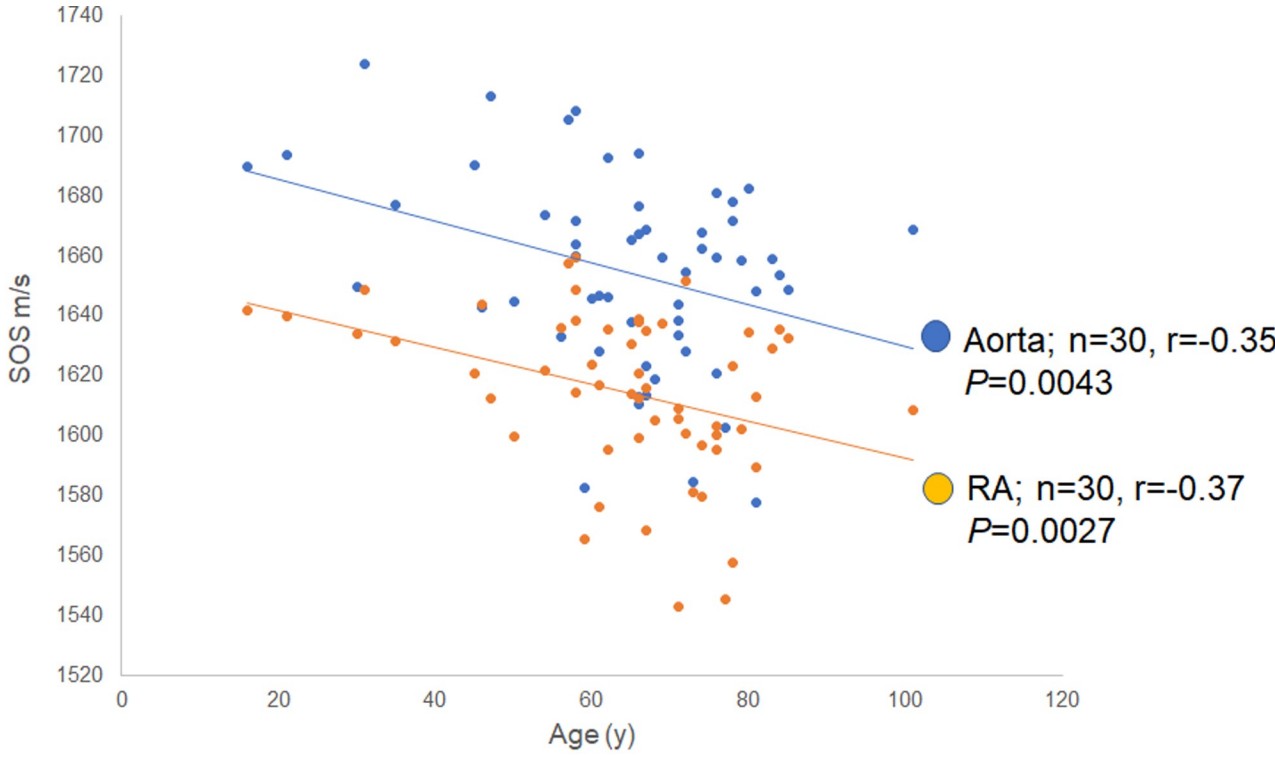

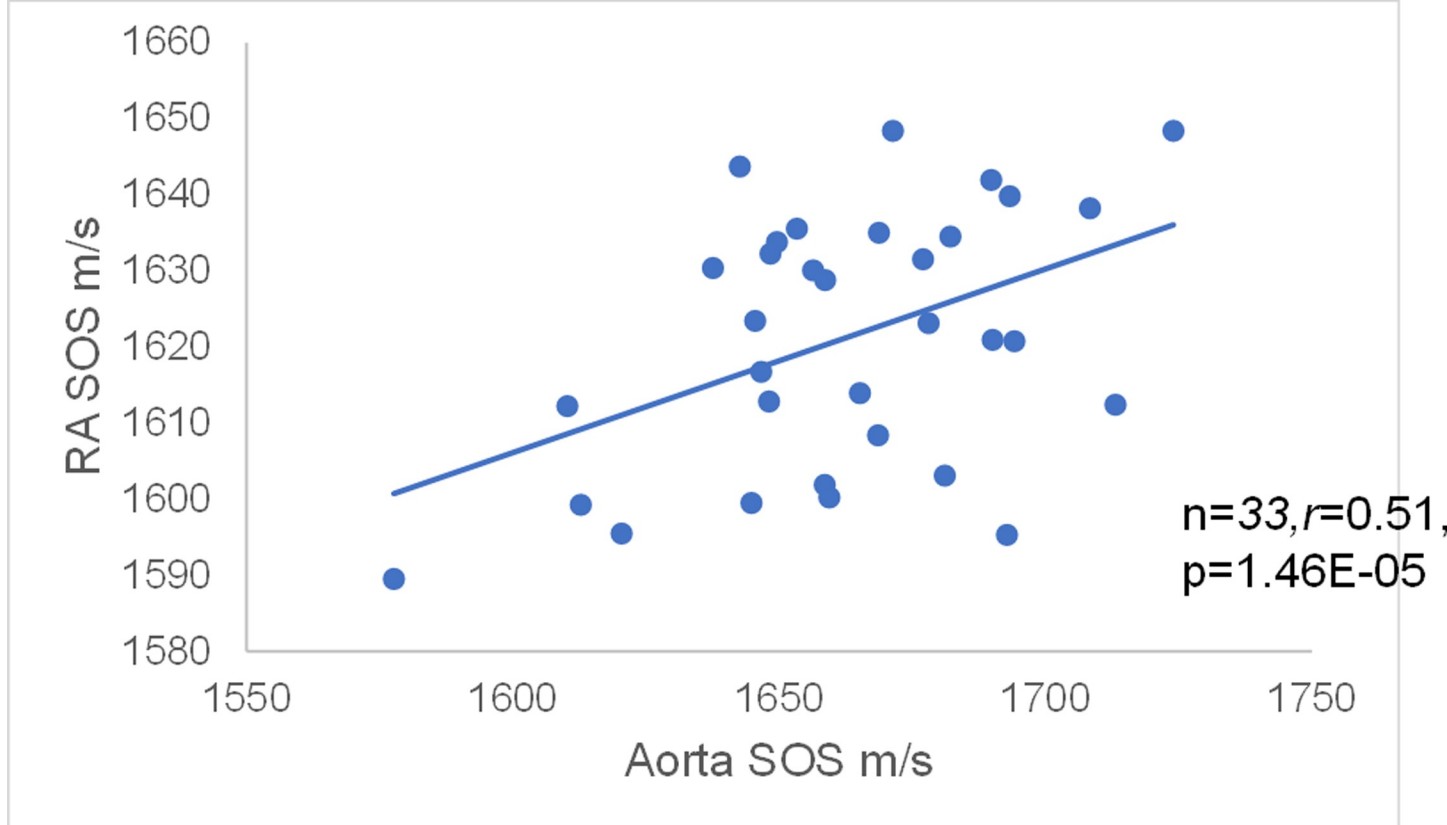

**Fig 7. Relationship of speed-of-sound (SOS) values of the renal artery (RA) and ascending aorta with aging.** (a) SOS values of tunica media of RAs and ascending aorta with aging. Both SOS values progressively decreased with aging. SOS values of the aorta decreased following those of RAs (n = 33, r = −0.35, P = 0.0043). (b)

Relationship between speed-of-sound (SOS) values of the renal artery (RA) and the aorta. SOS of the ascending aorta was positively correlated with that of the RA (n = 33, $r = 0.51$, $P = 0.0000146$).

When tissue is stressed, the strain initially increases rapidly due to the elimination of free fluid which acts as a damper to delay the stress-strain response.

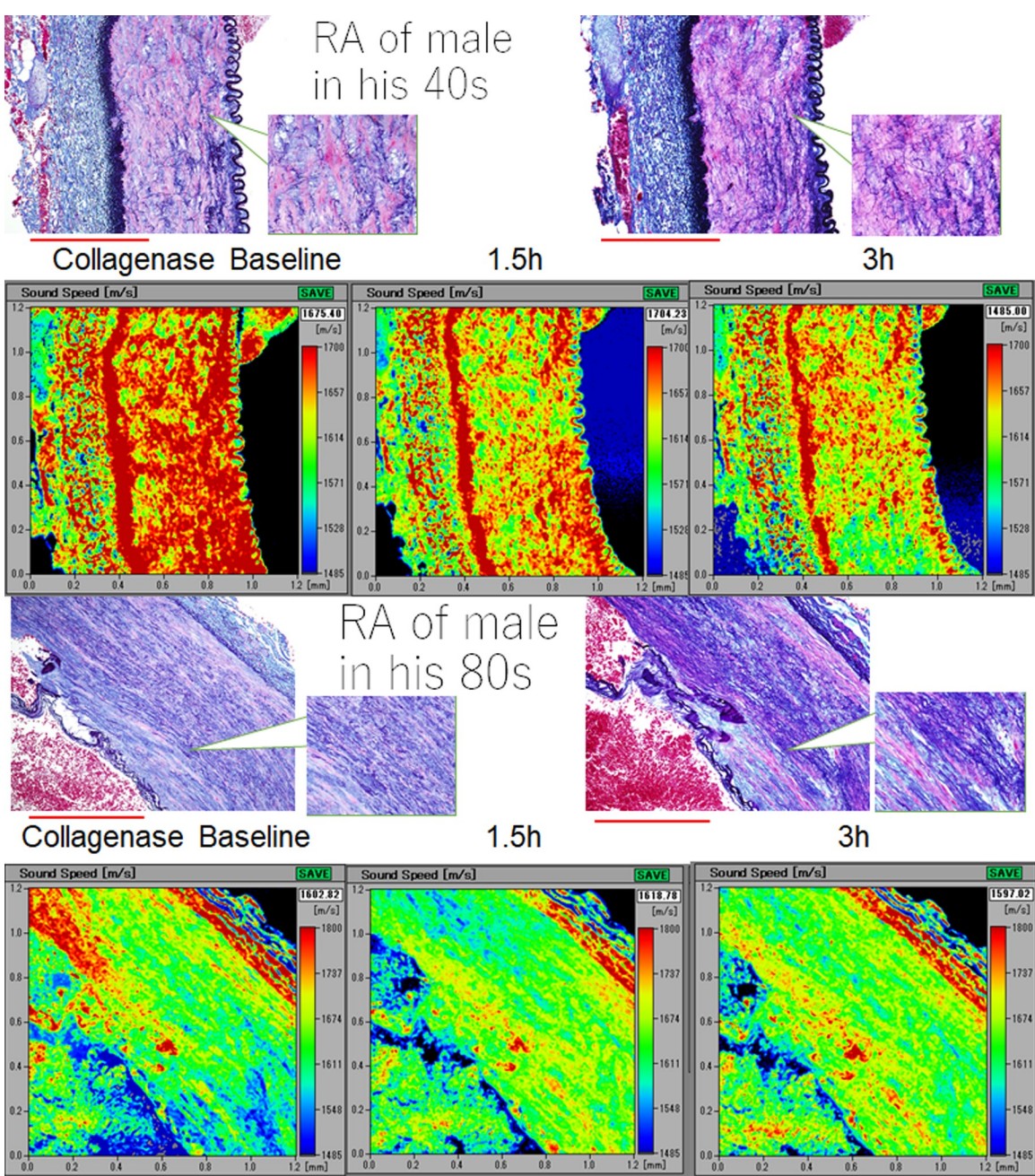

**Fig 8. Difference in sensitivity to collagenase digestion between young and old renal arteries (RAs).** Speed-of-sound (SOS) values before digestion were higher in the tunica media of young renal arteries (RAs) than in old RAs, and they reduced rapidly after digestion. However, old RAs were rather resistant to enzymatic degradation. The corresponding LM images in Verhoeff's elastic and Masson's trichrome stain were shown on the SAM images (left; before digestion, right; 3 h after digestion). The scale bars represent 500 μm.

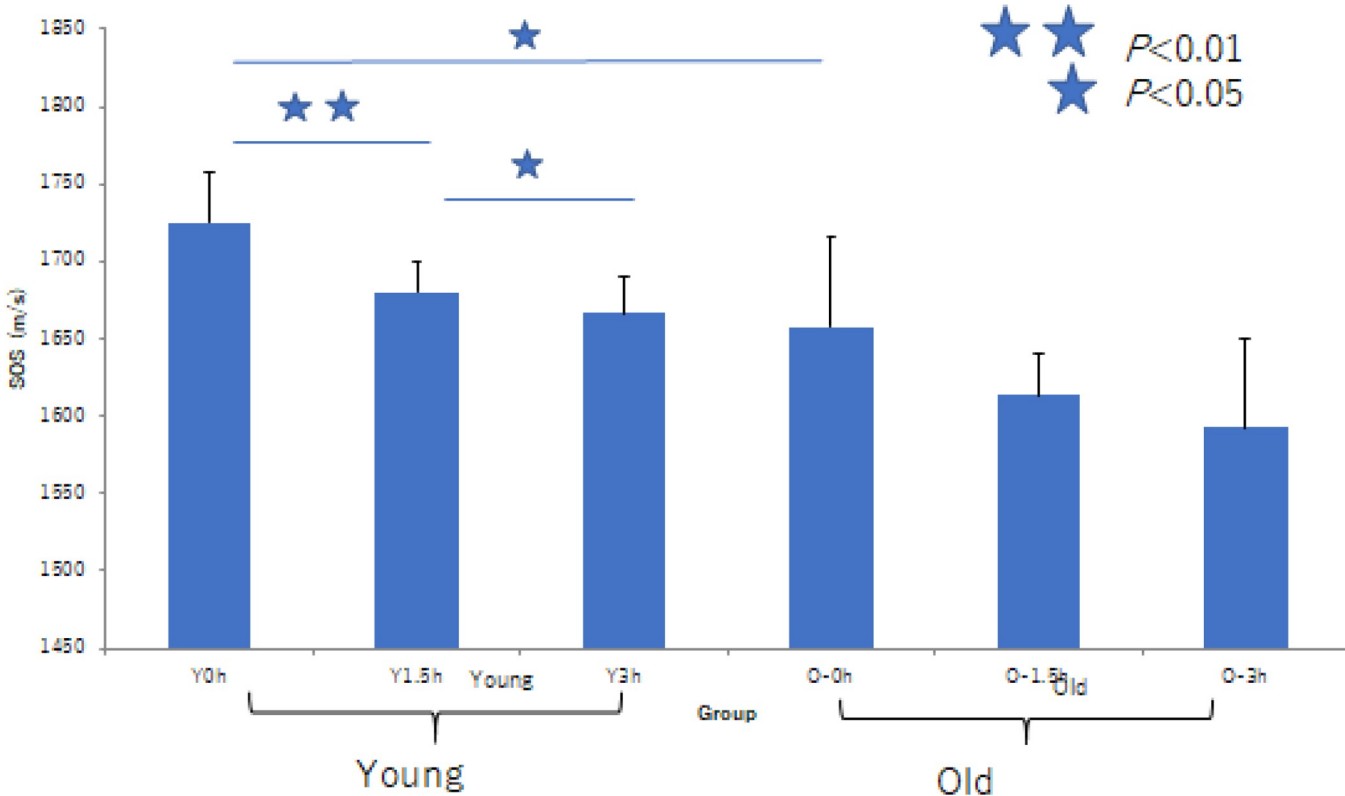

**Fig 9. Difference of susceptibility to collagenase digestion associated with aging.** Speed-of-sound (SOS) values before digestion were higher in the tunica media of young renal arteries (RAs) than in that of old RAs, and they reduced rapidly after digestion. However, old RAs were rather resistant to enzymatic degradation. The graph shows a significant decline in SOS of young RAs in comparison with old RAs. **$P<0.01$, *$P<0.05$.

Soft tissues consist of many materials and their mechanical properties cannot be described in terms of simple agglomerations of the simple materials, but by microscopic organizations. Published data on Young's modulus [17] and SOS values [18] of soft tissues corresponded well with its palpable elasticity and diseased state. Real SOS values reflect total properties of whole constituents and organization; therefore, SOS monitoring is useful for following the progression of structural and mechanical alteration with age.

Virtual histological images constructed from SOS values corresponded well with LM images and demonstrated alterations associated with aging. Older RAs became elliptical with hypertrophy of the tunica intima due to atherosclerosis. SOS of the tunica media progressively decreased with aging, which indicated mechanical weakness and was correlated with muscular atrophy and disappearance. Moreover, SOS values of EEL and IEL reduced with aging, indicating a loss of elasticity in old RAs. These results suggest that RAs stiffen with age mainly because of intimal atherosclerosis and accompanied degeneration of other layers with loss of muscle and elastic fibers. The hypertrophy of the tunica intima may compensate for the mechanical weakness of the outer layer to bear high BP.

Although arteries stiffen with age, the present study showed that SOS values lowered in proportion with age and BP. PWV and PP are the two significant indices of arterial stiffness [3] that typically increase with age [4,19]. PWV is assessed by measuring transit distance and transit time between two sites in arteries, such as carotid and femoral arteries. Pressure waveforms are simultaneously recorded by placing BP cuffs around the neck and upper thigh. Arterial distensibility, a measure of the artery's ability to expand and contract with cardiac pulsation and

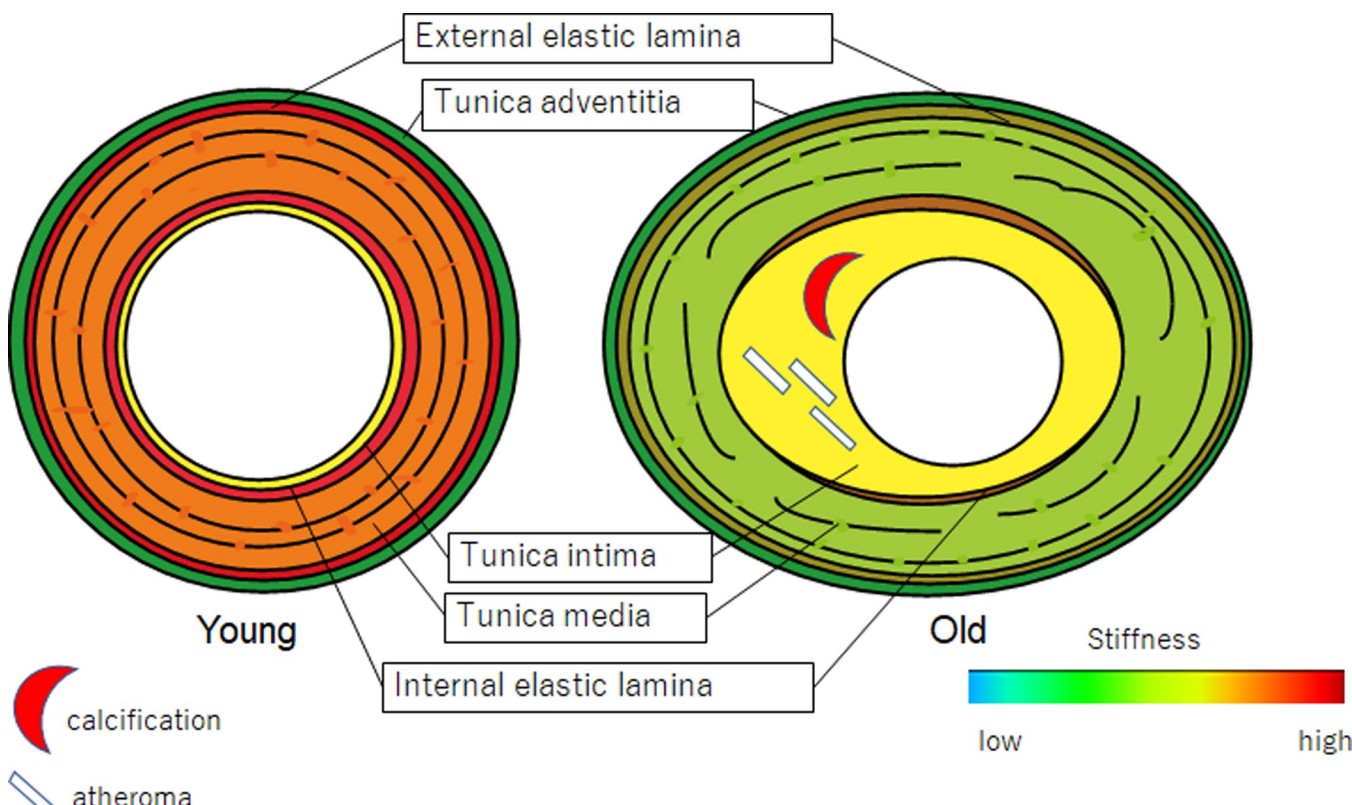

**Fig 10. Schematic image alterations of renal arteries (RAs) associated with aging.** The outer shape of the RAs changes from round to oval with inward and outward hypertrophy. The tunica intima stiffens with atherosclerosis, and the tunica media expands with a reduction in smooth muscle and splitting of elastic fibers.

relaxation, correlates with the degree of atherosclerosis, i.e., the intimal stiffness [20,21]. The present study reflected no intimal stiffness because tissue preparation procedures led to the loss of original intimal properties such as lipid accumulation and calcification. The discrepancy of the results depends on the areas assessed. PWV and SOS mainly calculate intimal and medial stiffness, respectively.

Some studies have already reported the relationship between SOS values and BP. Akhtar *et al.* stated that SOS had an inverse relationship with systolic and diastolic blood pressure in aortic biopsies [22]. Diabetic aorta—one of the causes of hypertension—showed reduced SOS in vessel walls, particularly in the interlamellar regions of the tunica media in an experimental rat model [23]. These regions corresponded to the extracellular matrix in which protease activity was increased in diabetic vessels. Fibrillin microfibrils, one of the extracellular matrix proteins, were significantly shorter in diabetic rats than in healthy controls. Reduced muscle fibers and microfibril fragmentation may cause mechanical weakness of the tunica media. Atherosclerotic intimal lesions reportedly cause increased SOS values [24].

Old RAs showed resistance to protease digestion in comparison with young RAs. Young RAs with concentrated muscles and regular elastic fibers showed more vulnerability to collagenase, as observed in thoracic aortae [13]. Old RAs with split elastic fibers and reduced smooth muscles, which had been replaced by collagen fibers, were already affected or modified to become resistant to protease digestion. Therefore, the extracellular matrix components of old RAs were maintained after digestion.

With aging, the tunica media lost its stiffness and its diameter enlarged; however, its thickness remained stable, which may be an adaptive response to high BP. This outward hypertrophic remodeling corresponds to that in animal models of normal aging [19]. Collagen fibers (via fibrosis) and extracellular matrix components fill the space among smooth muscles in old RAs. Computed tomographic angiography in adults revealed that mean RA diameter increased from the 10s to 50s, was rather stable up to 70s, and rapidly decreased after 80s in the Nigerian population [25]. This result showed that adult RAs usually become dilated with aging and finally reveal luminal stenosis via atherosclerosis, consistent with the present autopsy cases. The present research involved autopsy cases in which the patients had had chronic debilitating diseases. Therefore, degenerative changes in the tunica media of these cases were more severe than those observed in the tunica media of healthy adults.

SOS values in the ascending aorta were positively correlated with those in the RA (Fig 9B). Both vessels have a similar structure consisting of smooth muscles and elastic fibers [1]. Aortic pulse waves are conducted to RAs; therefore, RA pressure is always lower than the aortic pressure. It is reasonable that SOS values of RA are lower than those of aorta in all ages (Fig 9A).

In my previous study, SOS values in the thoracic aorta were negatively correlated with age [13]. Older aortae showed more significant degeneration of the tunica media with low SOS values. These aortae expressed specific extracellular matrix components to compensate for mechanical weakness. RAs with a similar decrease in SOS values associated with aging exhibited similar histology, probably following the same process. The resistance of old RAs to collagenase digestion signified that older RAs possessed more modification or bridging of proteins [15,26].

This study had several limitations. First, all samples were obtained from autopsy cases, of which a high percentage represented patients with neoplasms. Most patients with tumors had received numerous types of therapies that nutrition was in a poor state. This state of malnutrition might cause reduced lipid accumulation in the tunica intima. Prolonged formalin fixation might influence SOS measurement; however, Sasaki et al. [14] reported that the influence of formalin fixation on the acoustic properties of the healthy kidney was minimal. No significant change in acoustic parameters, including SOS, was found as seen in Figs 1 and 2. Second, location bias might influence the results. RAs with severe focal calcification could not be cut into flat sections; therefore, these hard portions were excluded from the study. Decalcification procedure by soaking in acid solution may influence SOS values. Third, the height and weight of the cadavers were not considered; therefore, the influence of body size on the findings was not assessed. Although the RA size was correlated with body size, renal function remained in the normal range and might have changed during aging; owing to this reason, raw data were used.

## Conclusions

This study revealed the utility of SAM observation, which simultaneously showed structural and mechanical information from a histological glass slide. It provided objective evidence of damage or degeneration in each portion of the RA. The fact that arteries become stiffer with aging originates from intimal atherosclerosis and not from medial degenerative changes. SAM investigation disclosed the association between aging-related structural and functional alterations.

## Supporting information

**S1 Table. Autopsy cases used in this study.**
(DOCX)

**S2 Table. SOS alteration among fresh-frozen in different fixation and FFPE sections.**
(DOCX)

**S3 Table. Relationship between age and speed-of-sound (SOS of the medial layer of renal artery.**
(DOCX)

**S4 Table. Speed-of-sound values of internal (IEL) and external (EEL) elastic laminae associated with aging.**
(DOCX)

**S5 Table. Mean blood pressure and speed-of-sound values of the tunica media of renal artery.**
(DOCX)

**S6 Table. Relationship between systolic (SBP) and diastolic (DBP) blood pressure and age.**
(DOCX)

**S7 Table. Thickness of each layer of the renal artery with aging.**
(DOCX)

**S8 Table. Length of medial inner and outer axes with age.**
(DOCX)

**S9 Table. Average speed of sound of renal and aortic medial layer.**
(DOCX)

**S10 Table. SOS alteration after collagenase.**
(DOCX)

**S1 File. SOS-images alteration of mouse artery among fresh-frozen sections in different fixations and formalin-fixed paraffin-embedded (FFPE) sections.**
(DOCX)

## Acknowledgments

The author thanks T. Moriki, Y. Egawa, Y. Kawabata, and N. Suzuki for their assistance in the preparation of the histological samples, Dr. K. Kobayashi (Honda Electronics) for his technical support and advice with SAM, and Enago (www.enago.jp) for the English language review.

## Author Contributions

**Conceptualization:** Katsutoshi Miura.

**Data curation:** Katsutoshi Miura.

**Funding acquisition:** Katsutoshi Miura.

**Investigation:** Katsutoshi Miura.

**Methodology:** Katsutoshi Miura.

**Resources:** Katsutoshi Miura.

**Writing – original draft:** Katsutoshi Miura.

**Writing – review & editing:** Katsutoshi Miura.

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
