## [Decision Letter · Decision Letter 0]

5 Aug 2020

PONE-D-20-16504

Tunica intima compensation for reduced stiffness of the tunica media in aging renal arteries as measured with scanning acoustic microscopy

PLOS ONE

Dear Dr. Miura,

Thank you for submitting your manuscript to PLOS ONE. After careful consideration, we feel that it has merit but does not fully meet PLOS ONE’s publication criteria as it currently stands. Therefore, we invite you to submit a revised version of the manuscript that addresses the points raised during the review process.

All the concerns raised from the reviewer listed below need to be addressed point-by-point. The mechanical property under different fixation need to be quantified in the revised manuscript.

We look forward to receiving your revised manuscript.

Kind regards,

Jun Yu, MD

Academic Editor

PLOS ONE

Journal Requirements:

Reviewers' comments:

Reviewer's Responses to Questions

**Comments to the Author**

1. Is the manuscript technically sound, and do the data support the conclusions?

Reviewer #1: Partly

2. Has the statistical analysis been performed appropriately and rigorously? 

Reviewer #1: Yes

3. Have the authors made all data underlying the findings in their manuscript fully available?

Reviewer #1: Yes

4. Is the manuscript presented in an intelligible fashion and written in standard English?

Reviewer #1: Yes

5. Review Comments to the Author

Reviewer #1: The paper describes a use of high spatial resolution (~4um) acoustic microscopy (AM) to evaluate mechanical properties changes in renal artery. While the approach is interesting and clearly is a great complement to the optical imaging and is compared with various staining methods, several points have to be fully addressed before the paper can be recommended for the publication in PLOS ONE.

1. The process of fixation, embedding and de-waxing can significantly change the mechanical properties of the sample introducing speed of sound (SOS) artefacts connected with eg selective changes in the areas with higher lipids concentration. These have to be quantified for the statistical analysis involved.

2. Line 125 compares the measurements of fixed samples with the "fresh ones". It is not clear how the latter were obtained, how the section was made, etc. It also might be illustrative to compare statistically the paraffin embedded and microtome sectioned samples with the cryotome processed samples.

3. Some protocols of dewaxing require ethanol step as xylene is practically not soluble in water. It is not clear how the authors omitted this step.

4. The AM measured SOS is not directly relevant to the flexural physiologically relevant mechanical properties of RA, with authors comment to the link between these expected.

6. PLOS authors have the option to publish the peer review history of their article (what does this mean?). If published, this will include your full peer review and any attached files.

Reviewer #1: No

---

## [Author Response · Author response to Decision Letter 0]

3 Sep 2020

Response to reviewer(s)

Reviewer #1: The paper describes a use of high spatial resolution (~4um) acoustic microscopy (AM) to evaluate mechanical properties changes in renal artery. While the approach is interesting and clearly is a great complement to the optical imaging and is compared with various staining methods, several points have to be fully addressed before the paper can be recommended for the publication in PLOS ONE.

1. The process of fixation, embedding and de-waxing can significantly change the mechanical properties of the sample introducing speed of sound (SOS) artefacts connected with eg selective changes in the areas with higher lipids concentration. These have to be quantified for the statistical analysis involved.

Answer

Thank you for your constructive comment. I have shown the comparative SOS data between fresh-frozen and FFPE sections with higher lipid area, as suggested.

Lanes 116–124, 180–196, Fig. 1(New), Fig. 2(New), and Table S1.

2. Line 125 compares the measurements of fixed samples with the "fresh ones". It is not clear how the latter were obtained, how the section was made, etc. It also might be illustrative to compare statistically the paraffin embedded and microtome sectioned samples with the cryotome processed samples.

Answer

Thank you for your comments. The fresh samples obtained through sectioning with cryotome were dried and fixed by soaking in 95% ethanol and 10% buffered formalin. The SOS images were captured during this time using the same section by removing the fixatives in distilled water at each time point to evaluate the effects of fixation. Moreover, the fresh tissues were fixed in 10% buffered formalin and embedded in paraffin. We compared SOS images of these paraffin-embedded sections with fresh cryostat sections.

I added the comparative images and SOS alteration of mouse arteries to the supporting file as follows:

S1 File

Changes in SOS images of mouse artery among fresh-frozen sections in different fixatives and formalin-fixed paraffin-embedded (FFPE) sections.

Materials and Methods

Fresh mouse arteries were frozen and sectioned with a cryotome. The sections were dried, fixed in 95% ethanol, and then soaked in 10% buffered formalin. The same section was used to compare the images after different fixation conditions. The residual frozen tissue cut with the cryotome were fixed in 10% buffered formalin for one day, embedded in paraffin, and sectioned with a microtome. The deparaffinized and frozen sections were soaked in distilled water and measured with SAM to compare their images.

Results

SOS images after 95% ethanol or 10% formalin fixation at different time points showed no remarkable changes (Figs. A and B, Table C). However, FFPE sections significantly increased SOS values on the arterial walls of the smooth muscle layer (P = 8.6E – 09)

Fig A. SOS images among fresh samples in different fixatives and formalin-fixed paraffin-embedded samples (FFPE). SOS of fresh-frozen section with different fixatives showed no remarkable changes. However, FFPE sections significantly increased SOS values on the arterial walls of the smooth muscle. The corresponding LM images in HE stain were shown at the same magnification.

Sample A

Sample B

 . 

Sample C

Form: formalin; FFPE: formalin-fixed paraffin-embedded; A, B, C: Different samples

Fig. B. Comparison of SOS values in different fixatives

Table C. SOS values (mean ± SD) among fresh samples in different fixatives and formalin-fixed paraffin-embedded samples. 

 EtOH20sec Et OH10m Form3m Form60m FFPE

A 1643.1±31.1 1628.5±29.4 1664.9±23.1 1674.4±43.8 1799.4±17.5

B 1643.7±30.5 1649.0±22.2 1646.3±23.1 1652.9±22.4 1781.5±19.7

C 1625.0±27.9 1636.6±13.8 1653.9±8.4 1663.8±17.0 1719.5±22.0

Form: formalin; FFPE: formalin-fixed paraffin-embedded; A, B, C: Different samples

. 

3. Some protocols of dewaxing require ethanol step as xylene is practically not soluble in water. It is not clear how the authors omitted this step.

Answer

I made a mistake by omitting the description of this step. I added this step to the revised manuscript, as suggested, which is shown below:

Line 115

Briefly, the 10-µm-thick specimen was dewaxed in xylene, soaked in ethanol at gradually decreasing concentrations, and washed in distilled water.

4. The AM measured SOS is not directly relevant to the flexural physiologically relevant mechanical properties of RA, with authors comment to the link between these expected.

Answer

Thank you for your important comment. The SOS values measured by SAM are not directly relevant to the flexural mechanical properties. However, SOS and soft tissue elasticity have a positive correlation, as shown in the following equation (Saijo 2009, Wells 2011):

C = (K/ρ) 1/2

Where c = speed of sound; K = elastic bulk modulus; and ρ = tissue density.

For a homogeneous isotropic solid material, the ratio of stress (force per unit area) /strain (proportional deformation) is a constant, called the modulus of elasticity. Young’s modulus (longitudinal elasticity), shear or torsion modulus (rigidity), and bulk modulus (volume elasticity) are commonly applied to compare the elasticity of biological tissues. 

When a material is stressed, its breadth may contract as its length extends. This is defined by a constant called Poisson’s ratio, which is given by

σ= (lateral contraction per unit breadth)/(longitudinal extension per unit length).

Poisson’s ratio is usually between 0.490 and 0.499 for soft tissues which are almost incompressible (Well 2011). 

There are 3 constitutive equations to define the relationship between 3 elastic modules.

G = E / (2(1 + σ))

σ = (E / 2G) – 1

K = E / (3(1 – 2σ))

Where G = shear modulus, E = Young’s modulus, σ= Poisson’s ratio, K = bulk modulus

Soft tissues, such as RAs, are not homogeneous isotropic materials but heterogenous anisotropic materials with elastic and viscous properties that show a variable stress-strain response. When tissue is stressed, the strain initially increases rapidly due to the elimination of free fluid which acts as a damper to delay the stress-strain response.

Soft tissues consist of many materials and their mechanical properties cannot be described in terms of simple agglomerations of the simple materials, but by microscopic organizations. Published data on Young’s modulus and SOS values of soft tissues corresponded well to its palpable elasticity and diseased state. Real SOS values reflect total properties of whole constituents and organization. Therefore, SOS monitoring is useful for following age-related progression of structural and mechanical changes.

I added the above sentences to the first paragraph in the Discussion, as suggested.

From Line 325

**According to the PLOS Data policy, I have provided all the data obtained in this investigation as supporting information.

---

## [Decision Letter · Decision Letter 1]

30 Sep 2020

Tunica intima compensation for reduced stiffness of the tunica media in aging renal arteries as measured with scanning acoustic microscopy

PONE-D-20-16504R1

Dear Dr. Miura,

We’re pleased to inform you that your manuscript has been judged scientifically suitable for publication and will be formally accepted for publication once it meets all outstanding technical requirements.

Kind regards,

Jun Yu, MD

Academic Editor

PLOS ONE

Additional Editor Comments (optional):

Reviewers' comments:

Reviewer's Responses to Questions

**Comments to the Author**

1. If the authors have adequately addressed your comments raised in a previous round of review and you feel that this manuscript is now acceptable for publication, you may indicate that here to bypass the “Comments to the Author” section, enter your conflict of interest statement in the “Confidential to Editor” section, and submit your "Accept" recommendation.

Reviewer #1: All comments have been addressed

2. Is the manuscript technically sound, and do the data support the conclusions?

Reviewer #1: Yes

3. Has the statistical analysis been performed appropriately and rigorously? 

Reviewer #1: Yes

4. Have the authors made all data underlying the findings in their manuscript fully available?

Reviewer #1: Yes

5. Is the manuscript presented in an intelligible fashion and written in standard English?

Reviewer #1: Yes

6. Review Comments to the Author

Reviewer #1: I appreciate the changes in the manuscript, in particular detailed validation of the fixation protocols and correlation of acoustic and rheological data. I can now recommend publication of the revised manuscript in PONE.

7. PLOS authors have the option to publish the peer review history of their article (what does this mean?). If published, this will include your full peer review and any attached files.

Reviewer #1: No

---

## [Editor Report · Acceptance letter]

7 Oct 2020

PONE-D-20-16504R1 

Tunica intima compensation for reduced stiffness of the tunica media in aging renal arteries as measured with scanning acoustic microscopy 

Dear Dr. Miura:

I'm pleased to inform you that your manuscript has been deemed suitable for publication in PLOS ONE. Congratulations! Your manuscript is now with our production department. 

Kind regards, 

on behalf of

Dr. Jun Yu 

Academic Editor

PLOS ONE